# Aptamers Enhance Oncolytic Viruses’ Antitumor Efficacy

**DOI:** 10.3390/pharmaceutics15010151

**Published:** 2022-12-31

**Authors:** Maya A. Dymova, Anna S. Kichkailo, Elena V. Kuligina, Vladimir A. Richter

**Affiliations:** 1Institute of Chemical Biology and Fundamental Medicine, Siberian Branch of the Russian Academy of Sciences, 8 Lavrentiev Avenue, 630090 Novosibirsk, Russia; 2Laboratory for Biomolecular and Medical Technologies, Krasnoyarsk State Medical University, 1 Partizana Zheleznyaka, 660022 Krasnoyarsk, Russia; 3Laboratory for Digital Controlled Drugs and Theranostics, Federal Research Center “Krasnoyarsk Science Center of the Siberian Branch of the Russian Academy of Science”, 660036 Krasnoyarsk, Russia

**Keywords:** oncolytic viruses, aptamer, cancer, neutralizing antibodies, cryoprotection, aggregation

## Abstract

Oncolytic viruses are highly promising for cancer treatment because they target and lyse tumor cells. These genetically engineered vectors introduce therapeutic or immunostimulatory genes into the tumor. However, viral therapy is not always safe and effective. Several problems are related to oncolytic viruses’ targeted delivery to the tumor and immune system neutralization in the bloodstream. Cryoprotection and preventing viral particles from aggregating during storage are other critical issues. Aptamers, short RNA, or DNA oligonucleotides may help to crawl through this bottleneck. They are not immunogenic, are easily synthesized, can be chemically modified, and are not very demanding in storage conditions. It is possible to select an aptamer that specifically binds to any target cell, oncolytic virus, or molecule using the SELEX technology. This review comprehensively highlights the most important research and methodological approaches related to oncolytic viruses and nucleic acid aptamers. Here, we also analyze possible future research directions for combining these two methodologies to improve the effectiveness of cancer virotherapy.

## 1. Introduction

According to GLOBOCAN 2020, cancer ranks second among all diseases in terms of mortality (9.84 million deaths from all types of cancer without non-melanoma skin cancer), second only to coronary heart disease [1]. Cancer epidemiology is not optimistic despite advances in new treatment and diagnostic approaches. Newly diagnosed cases are expected to reach 28.4 million by 2040. Therefore, developing new diagnostics and targeted cancer drugs remains an important and actual area of biotechnology, genetic engineering, and synthetic biology [2].

Today, there is a wide variety of platforms for creating anticancer drugs; in this review, we consider only two—oncolytic viruses and nucleic acid aptamers. Oncolytic viruses (OVs) represent a very auspicious direction for tumor treatment as they target cancer cells and carry therapeutic genes, the expression of which causes cytotoxic effects [3]. Aptamers have gained significant interest in biomedical science and pharmaceutics as perspective agents for targeted drug delivery. Aptamers have the potential to recognize and bind specifically to their targets in vitro and in vivo for successful therapeutic outcomes with minimal cytotoxic effects on surrounding normal cells. This review highlights the most crucial scientific achievements and methodological approaches for successfully using DNA or RNA aptamers and oncolytic viruses. Possible future research directions combining these nucleic acid ligands with virotherapy are discussed. Aptamers can effectively shield viruses from neutralizing antibodies (nAbs) and deliver them to tumor cells. Aptamers demonstrate cryoprotective properties and prevent viral particles from aggregating.

## 2. Oncolytic Viruses

The possibility of a practical antitumor effect of the virus was revealed in the middle of the twentieth century. We have come a long way from the idea to the development of oncolytic viral drugs approved for some clinical cancer treatment [4]. As there are a wide variety of viruses, it creates prerequisites for excellent prospects for personalized medicine. Several viruses including adenovirus, herpes simplex virus, Coxsackievirus, picornavirus, reovirus, Newcastle disease virus, vaccinia virus, vesicular stomatitis virus, and measles virus are being investigated to treat cancer because they infect, replicate, and lyse tumor cells, specifically inside cancer cells, without affecting normal cells and tissue counterparts [5,6]. It is also possible to deliver therapeutic or immunostimulating transgenes into a viral vector for expression at the tumor site. This approach is possible because, generally, cancer cells lack innate immune defense mechanisms against viruses involving the interferon system and are susceptible to viral infection [7]. As a rule, tumor cells have increased sensitivity to viral infection as they lose the ability to produce antiviral interferons and do not respond to exogenous interferons produced by normal cells. In turn, interferons could trigger various antitumor, proapoptotic mechanisms at the transcriptional and translational levels. Currently, virotherapy is used both as monotherapy or in combination with radiation, chemotherapy, immunotherapy, or surgical treatment [8].

Oncolytic viruses promote the destruction of tumor cells through a dual mechanism of action: direct cell lysis due to selective replication within neoplastic cells after the maximal production of progeny viruses, and the stimulation of a long-term antitumor immune response (Figure 1) [9,10]. Depending on the origin and type of cancer cell, the characteristics of the virus itself (type, dose etc.), the tumor microenvironment, and the host immune system, the contribution of these mechanisms may differ [11]. In normal cells, the antiviral response involves the activation of interferon receptors (IFNR) and Toll-like receptors (TLRs) [12]. TLRs are activated by binding to repetitive sequences, so-called pathogen-associated molecular patterns (PAMPs), characteristic of bacteria and viruses, and which are elements of the capsid, DNA, RNA, etc. Detection of viral nucleic acids by retinoic acid inducible gene 1 (RIG-1) also contributes to the antiviral response. A signaling cascade is then triggered through the nuclear factor κB (NF-κB) family and interferon regulatory factors (IRF 3, IRF 7), which activate specific transcription pathways and lead infected cells to apoptosis or necrosis [13]. IFNR activation induced by local IFN production promotes the JAK-STAT signaling pathway, resulting in upregulation of protein kinase R (PKR) and IRF7, which restrict the spread of the virus by binding to viral particles and trigger the type I interferon transcriptional network [14]. This cascade promotes apoptosis and cytokine production in infected cells, alerting the immune system to a viral infection. However, in cancer cells, this process is impaired. Downregulation of signaling proteins (RIG-1, IRF7, and IRF3) and key components of the type I IFN signaling pathway makes cancer cells more susceptible to viral replication.

Destruction of tumor cells by OVs results in the release of tumor-specific antigens (TSA), PAMPs, danger-associated molecular patterns (DAMPs), and cytokines (TNFα, IFNγ, and IL-12) [15]. These molecules can stimulate innate and adaptive immune responses by promoting the maturation of antigen-presenting cells and active effector T-cells that mediate antitumor immunity upon antigen recognition. It also enhances T-cell infiltration and repolarizes the immunosuppressive tumor microenvironment.

Viruses with oncolytic properties, high selectivity to tumor cells, and harmlessness for normal ones can be created by removing from the viral genome components involved in the suppression of protective mechanisms and introducing additional elements that increase the oncoselectivity of the virus. Thus, genes encoding proteins toxic to tumor cells (E3-11.6K and E4orf4 proteins) can be inserted into the viral genome [16,17]. This type of cell death is less preferred as it does not involve amplifying viral particles infecting neighboring tumor cells. Viruses can also carry immunostimulatory genes and, thus, trigger an antitumor immune response [11]. For example, recombinant viruses have been engineered to carry the following immunostimulatory genes (IL-2, IL-4, IL-12, and GM-CSF) as well as pro-apoptotic genes (TNFα, p53, Lactaptin, and TRAIL) [18,19]. Moreover, virotherapy with viral vectors carrying IL-12 or IL-12 with TNFα in combination with antibodies against PD1/PDL1 produced a significant antitumor effect [20,21]. The combination of virotherapy with immune checkpoint inhibitors therapy also provides excellent results [22]. In the future, it is worth paying attention not only to immunostimulatory genes, but also to the suppression of immunosuppressive genes expression, such as IL-10. The latter is an immunosuppressive cytokine that disrupts cytokine production, inhibits differentiation of dendritic cells, and stimulates myeloid-derived suppressor cells with potent immunosuppressive activity [23].

One of the reasons for the ineffective therapy is the presence of cancer stem cells (CSCs) with increased chemo- and radioresistance, the ability to self-renewing, differentiation, and metastasis [24]. Cancer stem cells have the property of relative quiescence, which allows them to avoid the influence of the drugs cytotoxic for proliferating cells (alkylating agents, inhibitors of topoisomerase, and DNA synthesis) [25]. The most commonly used CSC surface marker is prominin-1 (CD133), which is involved in the regulation of the mitogen-activated protein kinase (MAPK)/the serine-threonine kinase (Akt) signaling pathways. The MAPK/Akt pathway plays a key role in tumor survival, proliferation, and other hallmarks. Many successful studies showed the ability of the viruses to target CSC cell surface markers [26]. Thus, both conventional cancer cells and CSCs are destroyed, which reduces the likelihood of cancer recurrence and metastasis, increasing the patient’s survival.

The effectiveness of virotherapy may decrease due to the activation of some signaling pathways (JAK/STAT, NF-kB, and PI3K/AKT/mTOR) and increased expression of specific proteins (TGF-β receptor 1, CTLA-4, TIM-3, receptor tyrosine kinase, histone deacetylase, proteasome, tankirase, etc.). In addition, in the tumor microenvironment, there are a lot of specific cells (regulatory T-cells, M2 tumor-associated macrophages, and myeloid-derived suppressor cells). The effectiveness can be increased by supplementing OVs with small molecule treatment, epigenetic modulators (DNA methyltransferase and histone deacetylases), and miRNAs [27,28].

Recent clinical advances in the use of FDA-approved oncolytic viruses in cancer therapy have been associated with viruses such as the genetically modified herpes simplex virus—Talimogene Laherparepvec (T-VEC), for the local treatment of unresectable metastatic stage IIIB/C–IVM1a melanoma [29,30]. The first recombinant oncolytic adenovirus (Oncorin (also known as H101)) has been approved by the China Food and Drug Administration (CFDA) for the treatment of head and neck cancer in combination with chemotherapy [31]. ECHO-7 virus Rigvir was approved for the treatment of melanoma in Latvia, Georgia, and Armenia [32]. Teserpaturev/G47Δ (Delytact^®^), a third-generation (triple-mutated) recombinant oncolytic herpes simplex virus type 1, was approved for the treatment of malignant glioma in Japan [33]. Several clinical trials of the effectiveness of teserpaturev are being conducted with the following tumors—prostate cancer, recurrent olfactory neuroblastoma, and malignant pleural mesothelioma. According to the site (https://beta.clinicaltrials.gov/ accessed on 11 November 2022), by the keywords cancer and oncolytic virus, 169 clinical trial records of promising and potential viral candidates with varying recruiting statuses are found. The detailed reviews comprehensively analyzed the current advances in the development of oncolytic viruses, including those that undergo clinical and preclinical trials [34,35,36,37].

Despite the attractiveness of using viruses as anticancer drugs, several limitations should be considered. On the one hand, DNA viral vectors are highly stable, easy to construct, and introduce multiple therapeutic genes. They are considered the most adapted and clinically developed vectors. On the other hand, their use is limited in clinical practice due to the possibility of non-specific vector integration and insertional mutagenesis of any viral DNA vector [38,39]. The use of RNA viruses, despite their many attractive properties, is limited due to the complexity of genetic manipulations even with modern genetic technologies [40]. For some RNA viruses (arthropod-borne viruses), viral integration into host genome sequences depends on lineage-specific interactions and not on viral exposure [41].

Special attention when performing preclinical and clinical trials of OVs should be given to the evaluation of their toxicity, especially with high-dose intravenous administration, environmental shedding of viruses in urine, saliva, and feces, possible mutations and recombinations, and reversion to wild-type virus [42]. For example, for HadV, NDV, VV, mORV viruses, environmental shedding was observed; for VSV, CVA, PV, SVV, mORV viruses, the acquisition of mutations was detected.

The adaptive antiviral response and pre-existing vector immunity also prevent the widespread use of viral vectors such as adenoviral herpesvirus and poxvirus. However, under the condition of intratumoral administration of such a virus and in the presence of appropriate memory T cells, this can help overcome the immunosuppressive conditions of the tumor microenvironment (TME) and enhance an effective adaptive antitumor immune response [43,44].

In the organism, viruses can encounter several barriers [3]. Often, a viral drug is administered intratumorally, but due to the need to treat metastases, it becomes necessary to administer it systemically. However, replication-competent viral vectors are best administered for local intratumoral administration in order to reduce toxicity, while replication-deficient vectors having proposed tumor affinity can be used for systemic delivery [39,45]. For example, the extracellular enveloped virus, one of the two antigenically and structurally distinct infectious virions produced by the vaccinia viruses, is naturally resistant to antibody neutralization and complementarity and, therefore, could be used for systemic delivery [46]. Thus, current approaches should be aimed at increasing the tropism of the virus to tumor cells, reducing the immunosuppressive effect of TME, overcoming the problems associated with the sequestration of the oncolytic virus in the liver and spleen, evading neutralization by serum factors, and selectively increasing the vascular permeability to viruses. Several solutions can help to overcome the barriers that hamper efficient oncolytic virus delivery, including cell carriers; serotype switching; polymer shielding; mononuclear phagocytic system blockade; target endothelium; permeabilize vessels; the decrease in interstitial fluid pressure. Mainly, OVs can be deactivated by neutralizing antibodies and then rapidly withdrawn from circulation [47]. Increased circulation time can be achieved by polymer coating with polyethylene glycol (PEG) or poly-(N-(2-hydroxypropyl)methacrylamide) (HPMA), as well as by using pre-infected T cells as carriers of oncolytic viruses into the tumor site [48]. However, such approaches have their drawbacks because polymer coating reduces infectivity. Furthermore, the use of patient T-cells requires isolation, activation, and subsequent back-infusion into the patient, which makes it impractical for clinical use [49]. New aptamer-based approaches are currently being developed to increase the antitumor efficacy of OVs by shielding them from neutralizing antibodies [48,50].

## 3. Aptamers

Aptamers are artificial short single-stranded RNA or DNA molecules consisting of 15–100 nucleotides [51]. The term “aptamer” comes from the Latin word “aptus”, which means “to fit”. Aptamers have high specificity, selectivity, and binding affinity for their related targets (metal ions, chemical compounds, proteins, cells, and whole micro-organisms) with the equilibrium dissociation constants (KD) ranging from the nanomolar to picomolar range [52,53]. They can also form multiple three-dimensional (3D) structures, due to which aptamers can match a given protein target [54]. SELEX (Systematic Evolution of Ligands by EXponential Enrichment) technology can be used to find an aptamer for a target that does not yet have antibodies, or, if the antigen is toxic to the host animal, the aptamer can be created in vitro (Figure 2) [55].

Traditionally, the SELEX process requires 10–20 rounds of selection to obtain enriched pools of specific DNA/RNA aptamers, after which they are tested using standard biochemical methods [56,57]. When comparing RNA and DNA aptamers, it can be noted that DNA aptamers can be obtained by a more straightforward enrichment procedure and at a lower cost. They are characterized by better stability; however, using RNA aptamers, a higher level of structural diversity can be achieved. Aptamers can be chemically modified by incorporating various modified bases during synthesis. They are stable over a range of ionic conditions, pH, and temperature. Unlike other nucleic acid molecular probes, aptamers interact with and bind to their targets through structural recognition, which is why they are also called “chemical antibodies”. Their interaction with the target is similar to the mechanism of the antibodies. It can be characterized as hydrophobic, electrostatic, hydrogen bonds, and van der Waals interactions, as well as shape complementarity. At the same time, the developed monoclonal and polyclonal antibodies sometimes show only limited binding to the target. Aptamers have a high binding specificity to the target; accordingly, it is possible to isolate similar targets the antibodies cannot distinguish. Their small size makes it easier to penetrate tissues, for example, improving histochemical staining and SERS signal intensity. The low bioavailability, high cost, and long time required to develop and manufacture antibodies necessitate the rapid development of drugs that selectively bind to a desired epitope of a target protein of known sequence. In turn, the production of aptamers is much cheaper than the production of humanized antibodies, especially given the current progress in microfluidic systems for selecting aptamers [53,58]. The efficiency of intracellular delivery of an aptamer-based agent will depend in part on the recycling properties of their target and the induction of the receptor-mediated internalization when linking the aptamer to the surface marker. In addition, the intracellular routing of aptamers will be affected by the number of biomarkers on the cell surface, the macroscopic characteristics of the delivered aptamer conjugate (size and nature of the cargo), and the dominant endocytic pathways present in a given tumor cell type.

Aptamers act as high-affinity ligands and may be potential antagonists of disease-associated proteins. Aptamers have many advantages, such as low toxicity, easy tumor penetration, and the ability to cross the blood–brain barrier (BBB). It has been shown that a fusion aptamer specific for the epithelial cell adhesion molecule (EPCAM) and the transferrin receptor (TfR) can cross the BBB. Moreover, by loading such a bifunctional aptamer (EPCAM-TfR) with doxorubicin (DOX), it is possible to transcytose the endothelial cells of the BBB and deliver a payload for the treatment of EPCAM-positive brain metastases [59]. Aptamers have great potential for use as theranostic—diagnostic and therapeutic—tools due to their ability to diffuse through the entire tumor area in the intracranial cavity [60]. It is possible to design aptamers targeting specific cell surface biomarkers with high affinity and specificity by a combinatorial SELEX procedure [61,62]. Several variants of SELEX optimized the aptamer selection a while ago based on the proposed in silico analysis. A combination of docking and classical molecular dynamics simulations is often used to sample possible aptamer-target binding poses. Several computational methods were used to predict the binding energies and interactions in aptamer–target complexes: interaction analysis, docking, and molecular dynamics for free energy calculation. Recently, an approach “Structure and Interaction Based Drug Design (SIBDD)” has been developed to rationally design aptamers based on their structure and interaction with the target, using classical molecular dynamics simulation data combined with quantum chemical interaction energies [63]. SIBDD is a powerful combination of computational screening simulations and directed mutagenesis with experimental validation at each cycle to obtain oligonucleotides with high affinity and selectivity. This approach makes it possible to design aptamers with limited information about the target; this is relevant, especially for the dangerous boost-mutating infections (SARS-CoV-2) when access to samples is minimal. Thus, the first rounds of SIBDD are performed based on the computational folding results obtained from the target’s primary sequence. This methodology quickly adapts aptamers to various pathogen mutations.

Currently, only one aptamer (Pegaptanib, marketed as Macugen, Eyetech Pharmaceuticals/Pfizer) is approved by the FDA for use in clinical practice to treat age-related macular degeneration [64]. According to the site (https://beta.clinicaltrials.gov/ accessed on 11 November 2022), there are only eight records of clinical trials of potential aptamer candidates with varying recruitment statuses; the search was carried out using the keywords cancer and aptamer. Comprehensive reviews have recently been published listing and detailing aptamers in clinical and preclinical trials [65,66,67,68]. Very promising nucleolin-targeting aptamer AS1411 can reduce tumor cell growth and induce apoptosis in cancer cells [69]. Spiegelmer NOX-A12, targeting CXCL12, interferes with the migration and drug resistance of chronic lymphocytic leukemia (CLL), inhibits chemotaxis, and sensitizes CLL cells to cytotoxic drugs [70]. The aptamer to vimentin, NAS-24, delivered into the cancer cell by the natural polysaccharide arabinogalactan, induced apoptosis of mouse ascites adenocarcinoma cells in vitro and in vivo [71]. Adenocarcinoma growth was inhibited when a mixture of arabinogalactan and NAS-24 was injected intraperitoneally into the tumor-bearing mice. Thus, although aptamers have great underestimated potential and continue to be explored for various in vitro and in vivo applications such as cell imaging, drug delivery, anticancer therapy, and immunotherapy, further development is required in terms of in vivo stability.

## 4. Applications of Aptamers

### 4.1. Use of the Aptamers as Drug Delivery Vehicles

Therapeutic agents can be attached via non-covalent or covalent bonds to the nucleic acid aptamer, creating an aptamer drug conjugation (ApDC). Aptamer-conjugated doxorubicin (Dox), a widely used cancer chemotherapeutic agent, has been shown to be more therapeutically effective than Dox alone [72]. Another example of ApDC was developed by conjugating aptamer AS-42 specific to heat-shock proteins Hspa8 and Hsp90ab1 with Cis-dichlorodiammineplatinum and Siberian larch arabinogalactan. This cisplatin-arabinogalactan-aptamer demonstrated a cytotoxicity effect against Ehrlich’s carcinoma [73]. Similarly, aptamer-siRNA chimeras injected intratumorally effectively targeted prostate-specific membrane antigens (PSMA) in vivo [74]. The aptamer-miRNA chimeras were created on the base of the RNA aptamer recognizing the tyrosine kinase receptor Axl (GL21.T) and miR-137 or miR-212, and PDGFRβ aptamer conjugated with the antimiR-10b. These chimeras were transported across the BBB model in vitro, which may be promising for a new combinational therapy of glioblastoma [75].

Unfortunately, aptamers have a short circulating half-life, which reduces their effectiveness in vivo as drugs or delivery agents [52,76,77]. A polymer encapsulation can protect the drug and aptamer from the physiological environment without compromising their biological activity and biophysical properties and ensure the controlled release of active agents at optimal doses with the appropriate kinetics. The polymer coating can result in a reproducible and predictable drug release over an extended time. Thus, the therapeutic effects of drugs with a short half-life can be extended with the enhanced therapeutic effect and minimal side-effects. This strategy improves patient compliance by reducing drug wastage, the frequency of therapeutic procedures, and drug dosages. In particular, the aptamer can be conjugated with the functional groups of polylactic acid-polyethylene glycol (PLA-PEG) or polylactic-co-glycolic acid-polyethylene glycol (PLGA-PEG), where each group plays its own functional role: an aptamer as a targeting molecule, PLA or PLGA helps encapsulate and control drug release, while PEG increases the half-life of the resulting bioconjugate from the circulation [78]. Such aptamer–polymer chimeras improve the specificity of drug delivery by targeting various surface biomarkers, including the epithelial cell adhesion protein molecule (EpCAM) on the surface of breast adenocarcinoma and colon cancer cells [79,80]; nucleolin on acute myeloid leukemia, gliomas, kidney tumors, etc. [81,82,83,84,85]; PSMA on the surface of prostate tumor cells [86], and mucin-1 on lung epithelial cancer cells [87,88]. The increased biodegradability and biocompatibility of copolymers or liposomes open up excellent opportunities for their conjugation with aptamers for the targeted drug delivery [52].

Aptamer-conjugated bacteria (ApCB) should also be mentioned as one example of using aptamers as drug delivery vehicles. In 2021, Geng et al. proposed the conjugation of tumor-specific aptamers using a one-step amidation procedure with an attenuated Salmonella Typhimurium VNP20009 (VNP) bacterial surface [89]. The use of ApCB in 4T1- and H22-tumor-bearing mice showed increased antitumor efficacy with enhanced immune responses in the tumor.

### 4.2. The Use of the Aptamers for Diagnostics and Detection of Tumor Cells

Early cancer diagnosis before symptoms appears critical to treating the disease, increasing the patient’s chances of recovery. Unfortunately, low concentrations of biomarkers are observed in the early stages, so traditional diagnostic methods such as computed tomography, magnetic resonance imaging, immunohistochemistry, and serum assays for cancer biomarkers may not be effective. The ability of aptamers to specifically recognize and bind with high affinity to even a relatively small number of their targets (biomarkers) has made them a convenient and promising tool for the early detection and imaging of cancer [90,91]. Oncogenesis often promotes increased expression of biomarkers and/or cell surface receptors, as they trigger many pro-oncogenic signaling pathways, making them attractive for disease treatment. Cell surface macromolecules are involved in many biological processes, such as cell signaling, adhesion and migration, intercellular interactions, and communication between the intra- and extracellular environment. Most anticancer drugs target cell surface biomarkers. Many aptamers have been developed using SELEX protein and cell technologies to target cell surface biomarkers. These aptamers have been extensively studied as theranostic molecules in hematological malignancies; lung, liver, breast, ovarian, brain, colorectal, and pancreatic cancers; for identifying and characterizing CSCs [92]. Aptamers have been developed to target cancer stem cells AC133 epitope and CD133 protein. These aptamers can be internalized, which makes these aptamers an attractive tool for specific targeted drug delivery [93]. Using conjugates of nucleolin aptamers (AS 1411), a susceptible and straightforward colorimetric method using UV-vis spectrometry was developed to detect MCF-7 breast cancer cells [91]. Based on the DNA aptamer LC-18, an electrochemical aptasensor for diagnosing lung cancer in human blood was developed [94]. Deep learning and computer simulation of an array of experimental data were used to process the signals of the electrochemical aptasensor for early lung cancer diagnosis. Combining cyclic voltammogram characteristics made it possible to distinguish samples from lung cancer patients and healthy candidates. Several reviews focus on using aptamers to bind and visualize tumor cells and provide detailed descriptions of many successful examples [51,95,96].

### 4.3. Aptamers Used to Increase the Antitumor Efficacy of an Oncolytic Virus

#### 4.3.1. Use of the Aptamers to Shield the Oncolytic Virus from the Binding by Neutralizing Antibodies

Repeated administration of oncolytic viruses induces neutralizing antibodies (nAbs) production, decreasing the antitumor effect [97]. At the proof-of-concept level, novel methods for OVs shielding from nAbs have been proposed [49,50,98]. The antigen-binding fragment (Fab) region of the rabbit anti-vesicular stomatitis virus (VSV) neutralizing polyclonal antibodies was chosen as the target for aptamer selection. Such aptamers could inhibit the interaction between VSV and nAbs, thereby maintaining the anticancer effect of the virus. The evaluation of binding was performed using flow cytometry and electrochemical sensing. In particular, the degree of shielding (DoS) was measured by a bifunctional electrochemical immunosensor and consisted of more than 50% for several clones. A bifunctional electrochemical (cyclic voltammetry and electrochemical impedance spectroscopy) immunosensor confirmed the shielding effect of aptamers and made it possible to quantify OVs with high sensitivity, selectivity, low cost, and short analysis time compared to traditional methods such as plaque analysis, RT-PCR, ELISA, and optical sensors. A plaque-forming assay demonstrated that aptamers with high DoS retained biological activity. This approach (aptamer-mediated neutralizing antibody shielding (AptaNAS)) improves the survival of the oncolytic virus, preserves the efficiency of cancer cell infection in the presence of nAb, and improves the effectiveness of antitumor treatment. The developed multivalent aptamers further reduced virus aggregation, increasing infectivity and stability in human blood serum [50].

In addition, aptamer clones bind to immobilized antibodies, thus mimicking the binding to the antibody on the surface of B-lymphocytes. Shielding B cell receptors with injected aptamers can prevent B cell proliferation and, thus, inhibit subsequent production of neutralizing antibodies against the virus. It was shown that aptamer pools had a high protective effect on the cell, which can be explained by the fact that anti-VSV-nAbs are polyclonal. This pool can block various paratopes through different clones of DNA aptamers.

#### 4.3.2. Aptamers Change the Binding Specificity of Viruses

In 2009, the fundamental possibility of conjugating virus-like particles (VLPs) with the aptamers (VLP-aptamer) was shown for the first time. VLP-aptamers are natural nanomaterials with multifunctional properties that can be used for drug delivery [99]. Sixty DNA aptamer molecules specific to the tyrosine kinase receptor on Jurkat T cells were attached to each viral capsid. Microscopic analysis of cells incubated with fluorescently labeled aptamers attached to the viral capsid showed that VLPs underwent endocytosis and were transferred to lysosomes for the subsequent degradation. Such VLP-aptamers can be used for drug delivery of acid-labile prodrugs preferentially released upon lysosomal acidification. A general strategy has been developed to create natural packages and vehicles for the delivery of nanodrugs using virus-like particles based on the bacteriophage Qβ coat protein [100]. Thus, aptamers can be protected from degradation by VLP encapsulation. The release rate of drug payload can also be regulated by VLP channel sizes to avoid the use of high doses of drugs. Vice versa, virus-like particles represent a breakthrough in the delivery of aptamers to a target site. VLPs can improve the pharmacokinetics and biodistribution of the aptamers and the drug release pattern [101].

In the works mentioned above [49,50,98], DNA-aptamer selection against VSVs was also carried out. It was shown that the combination of anti-VSV and anti-nAbs aptamers resulted in increased infectivity, indicating a synergistic mechanism. To intensify the efficiency of these aptamers, dimeric and tetrameric aptamers linked by an oligonucleotide bridge were designed, the combination of which increased the infectivity in a greater extent.

COVID-19 is receiving a new round of viral aptamer research. These nucleic acid ligands can be a tool to inhibit or detect coronavirus. The gold nanoparticle spherical cocktail neutralizing aptamer (SNAP) was developed to block the interaction between the host functional receptor angiotensin-converting enzyme 2 (ACE2) and receptor-binding domain (RBD) of SARS-CoV-2 [102]. Using SELEX and molecular docking, the CoV2-6 aptamer has been identified and applied to prevent, compete, and replace ACE2 binding to the spike protein RBD, inhibiting SARS-CoV-2 infection [103]. A promising tool for diagnosing COVID-19 was elaborated based on aptamers targeting the spike (S) protein of SARS-CoV-2, which were immobilized on Au nanoparticles [104]. The S-protein was detected by electrochemical impedance spectroscopy.

#### 4.3.3. Aptamers as Cryoprotectors of Oncolytic Viruses and Aggregation Inhibitors

Viral vectors used in vaccination programs, and for genetic engineering for the development of antitumor drugs require stabilization at low temperatures. Viruses are temperature-sensitive; therefore, they must be stored frozen. Thus, delivery of the active viruses depends on the cold chain, a distribution network designed to maintain optimally low temperatures during transportation and storage. Several agents, such as metal ions, albumin, gelatin, and poly(ethylene glycol), have been used to increase the resistance of the viruses to low temperatures [105,106,107]. However, these compounds are characterized by low cryoprotection efficiency or high cellular toxicity. Thus, the development of cryoprotectants for viral vectors remains to be an urgent problem. Natural saccharides have recently attracted considerable attention due to their cryoprotective properties. These molecules can be used in media compositions to maintain vaccine quality and store virions and VLPs [108]. The quadramer of anti-VSV aptamers has also been shown to protect viral activity during multiple freeze–thaw cycles; in particular, increasing virus infectivity by 1.4 logs after 60 freeze–thaw cycles in a plaque formation assay prevents virus aggregation [98]. The latter is necessary for the further infection process because aggregation can limit the spread of viral particles and prevents the establishment of a successful infection in the early stages [109]. It is worth noting that the quantification and behavior of viruses in the environment during virus aggregation will be difficult. In general, viral aggregates can form under the influence of certain salts and their concentrations in solution, cationic polymers, and suspended organic matter near the isoelectric point of the virus [110].

We summarize all the information provided in Section 4.3. Aptamers increased the antitumor efficacy of an oncolytic virus, in Table 1.

## 5. Conclusions

Aptamers, an alternative class of targeting molecules, have exceptional characteristics that can be used to increase the effectiveness of virotherapy. They are designed to selectively bind to various targets, including proteins, cells, tissues, and microorganisms. There is the possibility of their chemical modification, universal regulation of their functional activity, scalable and cheap production, and long-term storage. They can be compared with the established monoclonal antibodies and used for targeted therapy or diagnostics in clinical practice and research studies. Oncolytic viruses have great potential for cancer virotherapy as they selectively lyse tumor cells. Normal cells and tissue counterparts are not affected by the oncolytic viruses. The release of tumor-associated antigens indirectly stimulates a long-term antitumor immune response, and various transgenes can modify them. The essential advantage of oncolytic viruses is their ability to lyse drug-resistant cancer stem cells. Both of the above platforms for developing cancer drugs can be improved independently. Thus, it is necessary to increase the short circulating half-life for the use of aptamers, which reduces their effectiveness in vivo as drugs or delivery agents by to their chemical modifications or conjugations with polymers, nanoparticles, etc. A platform based on viral vectors should solve the problem of targeting innate and adaptive antiviral immunity, increasing the expression of immunostimulatory cytokines in response to the release of tumor-associated antigens, conversely reducing the level of immunosuppression of TME. Here, we review the possibilities of using aptamers to improve oncolytic virotherapy (Figure 3).

Aptamers can be used to enhance the antitumor activity of oncolytic viruses through

(1)Shielding the virus from neutralizing antibodies;(2)Increasing the targeting and specificity of the viral particles;(3)Cryoprotective and anti-aggregative properties of the oligonucleotides bound to oncolytic viruses during transportation and storage.

Thus, it is possible to combine both technological platforms without comparing advantages and disadvantages, considering all the strengths of oncolytic viruses and aptamers to create targeted anti-cancer drugs and promote them in diagnostic and pharmaceutical markets. 

## Figures and Tables

**Figure 1 pharmaceutics-15-00151-f001:**
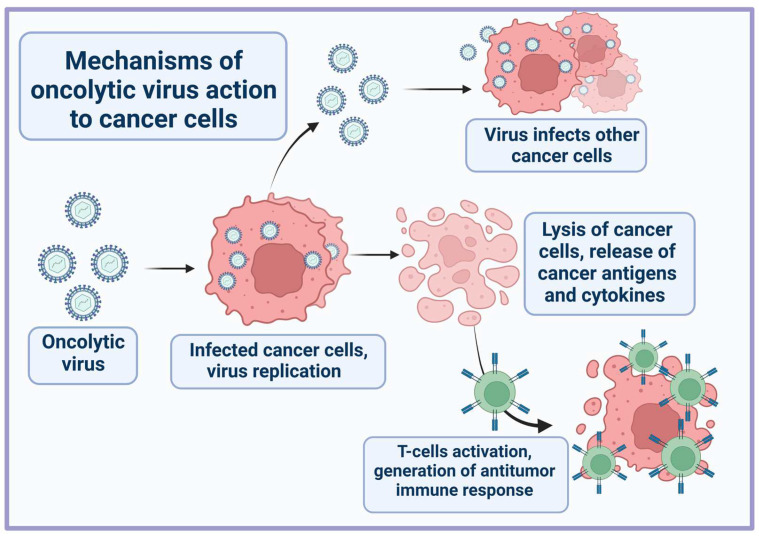
General scheme of oncolytic virus action on tumor cells.

**Figure 2 pharmaceutics-15-00151-f002:**
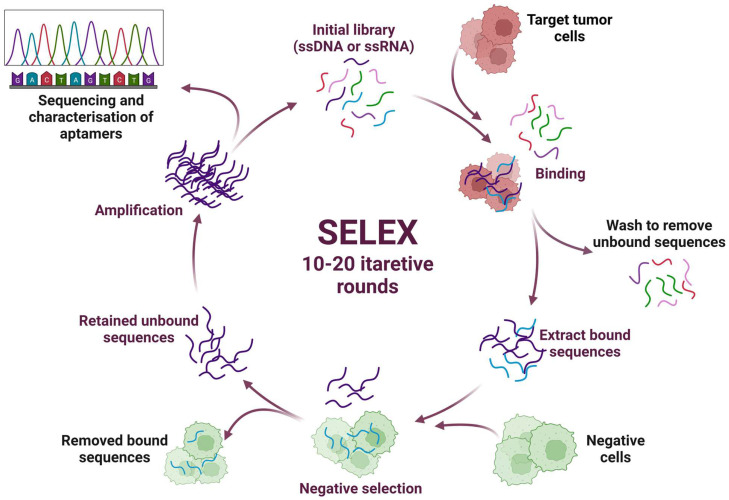
Schematic representation of the SELEX (systematic evolution of ligands by exponential enrichment) technology.

**Figure 3 pharmaceutics-15-00151-f003:**
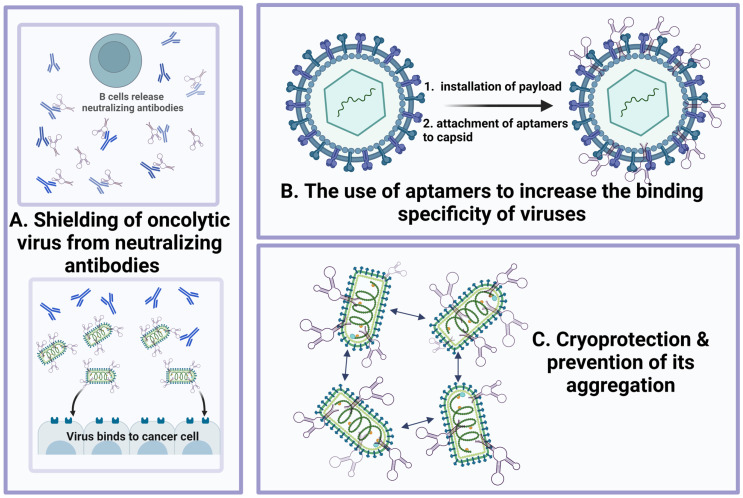
Aptamers increase the antitumor efficacy of oncolytic viruses through different mechanisms: (**A**) shielding oncolytic virus from the neutralizing antibodies (at the same time, aptamers can be specific both to Fab fragments of neutralizing antibodies and to enveloped proteins of the virus); (**B**) use of aptamers to increase the specificity of virus binding; (**C**) cryoprotection and prevention of virus aggregation.

**Table 1 pharmaceutics-15-00151-t001:** Aptamers increased the antitumor efficacy of the oncolytic viruses.

№	The Name of Aptamers	Virus	Targets	Mechanisms	References
1	41-nucleotide DNA aptamer	Bacteriophage MS2	Tyrosine kinase receptor on Jurkat T cells	cell binding, endocytosis, and translocation to lysosomes for degradation	[99]
2	RNA aptamer	VLPs from bacteriophage Qβ	heteroaryldihydropyrimidine structure	development of encapsulation technique	[100]
3	DNA aptamers	VSV	Fab of anti–VSV pAbs	shielding virus from nAbs	[49]
4	Tetramer of DNA aptamers	VSV	Fab of anti–VSV pAbs, VSVs	shielding the virus from nAbs, increasing infectivity	[50]
5	Quadramer of DNA aptamers	VSV	Fab of anti–VSV pAbs	Prevention of virus aggregation, protection against nAbs, cryoprotection	[98]

## Data Availability

Data sharing not applicable.

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
