# Peer review of "Aptamers Enhance Oncolytic Viruses’ Antitumor Efficacy"

_pharmaceutics, 2022, doi:10.3390/pharmaceutics15010151_

Round 1

Reviewer 1 Report

The authors have kindly submitted a manuscript “ Aptamers enhance oncolytic viruses’ antitumor efficacy.”

Upon review the authors make some interesting comments about the utility of Aptamers as therapeutic agents in the field of oncology.  

Please note the numerous grammatical errors makes this article difficult to review:

It has come a long way from the idea to the development of 48 drugs used in the clinic 

The mechanisms of cancer cells destruction can be different: 

This cell death type is less preferred, as it does not involve 81 the viral particles’ amplification which infects neighboring tumor cells. 

On the one hand, DNA viral vectors have high 120 stability, ease of construction and good ability to introduce several therapeutic genes, 121 they are considered the most adapted and clinically developed. 

They are characterized by greater 173 stability of the final aptamers, while using RNA aptamers a higher level of structural 

Aptamers act as high-affinity ligands and that can be potential antagonists of dis- 198 ease-associated proteins. Aptamers have a number of advantages such as low toxicity, 199 easy penetration into tumors and the ability to cross the blood-brain barrier (BBB). 

I can provide you a list of some organizations that provide these services should you feel you do not have the time to edit the manuscript yourself. 

https://www.enago.com 

https://www.biosciencewriters.com

https://www.lifescienceeditors.com

Please note the structure of the manuscript is not sufficient and the review is still in need of some organization.  For instance the authors should discuss more regarding aptamers as a therapy. So increase the information in section three, and shorten section two. 

For instance this statement should be used to link aptamer biology first to then oncolytic viruses as a therapy. 

Unfortunately, aptamers have a short circulating half-life, which reduces their ef- 226 fectiveness in vivo as drugs or delivery agents [40,52,53]. To ensure controlled release of 227 active agents at optimal doses with the appropriate release kinetics, a polymer encapsu- 228 lation can be used to protect the drug and aptamer from the physiological environment 229 without compromising their biological activity and biophysical properties 

Author Response

Dear Reviewers and Editors,

Thank you for giving us the opportunity to submit a revised draft of our manuscript titled “Aptamers enhance oncolytic viruses’ antitumor efficacy” to Pharmaceutics. We appreciate the time and effort that you and the reviewers have dedicated to providing your valuable feedback on our manuscript. We are grateful to the reviewers for their insightful comments on my paper. Please find below our response to reviewer’s comments.

Comments from Reviewer â„–1

The authors have kindly submitted a manuscript “Aptamers enhance oncolytic viruses’ antitumor efficacy”. Upon review the authors make some interesting comments about the utility of Aptamers as therapeutic agents in the field of oncology. Please note the numerous grammatical errors makes this article difficult to review:

  1. It has come a long way from the idea to the development of 48 drugs used in the clinic

Response: Here we once again express our gratitude to the reviewer for the identified flaw in the text. We change the phrase and reference from “It has come a long way from the idea to the development of drugs used in the clinic [3]” to “. A long way has been passed from the idea to the development of oncolytic viral drugs approved for some clinical cancer treatment [4]”. The reference has been replaced with a more recent one that accurately reflects the meaning of the sentence.

  1. The mechanisms of cancer cells destruction can be different:

Response: Corrected. We changed the phrase altogether.

  1. This cell death type is less preferred, as it does not involve the viral particles’ amplification which infects neighboring tumor cells.

Response: Corrected. We have changed the sentence to the another one “This type of cell death is less preferred since it does not involve amplifying viral particles infecting neighboring tumor cells”.

  1. On the one hand, DNA viral vectors have high stability, ease of construction and good ability to introduce several therapeutic genes, they are considered the most adapted and clinically developed.

Response: Corrected. We have changed the sentence to “On the one hand, DNA viral vectors are highly stable, easy to construct and introduce multiple therapeutic genes. They are considered the most adapted and clinically devel-oped vectors.”

  1. They are characterized by greater stability of the final aptamers, while using RNA aptamers a higher level of structural

Response: Corrected. We have changed the phrase to make it clearer “They are characterized by better stability, however, using RNA aptamers, a higher level of structural diversity can be achieved.”

  1. Aptamers act as high-affinity ligands and that can be potential antagonists of disease-associated proteins. Aptamers have a number of advantages such as low toxicity, easy penetration into tumors and the ability to cross the blood-brain barrier (BBB).

Response: Corrected. We changed the phrase to “Aptamers act as high-affinity ligands and may be potential antagonists of dis-ease-associated proteins. Aptamers have many advantages, such as low toxicity, easy tumor penetration, and the ability to cross the blood-brain barrier (BBB).”

  1. I can provide you a list of some organizations that provide these services should you feel you do not have the time to edit the manuscript yourself.

https://www.enago.com

https://www.biosciencewriters.com

https://www.lifescienceeditors.com

Response: Thank you for this suggestion. We reviewed the final text with a native English-speaking colleague with an individual Grammarly.com account.

  1. Please note the structure of the manuscript is not sufficient and the review is still in need of some organization. For instance, the authors should discuss more regarding aptamers as a therapy. So increase the information in section three, and shorten section two.

Response: Corrected. Based on the comments of all reviewers, the structure of the text has undergone significant changes, we have marked in yellow new sentences in the body of the article.

  1. For instance, this statement should be used to link aptamer biology first to then oncolytic viruses as a therapy.

Unfortunately, aptamers have a short circulating half-life, which reduces their effectiveness in vivo as drugs or delivery agents [40,52,53]. To ensure controlled release of active agents at optimal doses with the appropriate release kinetics, a polymer encapsulation can be used to protect the drug and aptamer from the physiological environment without compromising their biological activity and biophysical properties

Response: We agree with the reviewer. We have moved this text to the section 4.1. Use of the aptamers as drug delivery vehicles.

We highlighted (yellow highlighted text) all changes made when revising the manuscript to make it easier for the Editors to give a prompt decision on manuscript. We thank the editors for considering our work for publication.

Yours faithfully,

Dr. Maya Dymova PhD, Senior researcher

Laboratory of Biotechnology

Institute of Chemical Biology and Fundamental Medicine SB RAS

(ICHBFM SB RAS) Russian Federation, 630090, Novosibirsk, Lavrentjev av., 8

tel: 00 7 9134764012

Reviewer 2 Report

The review by Dymova A. et al. describes the current progress with the use of aptamers for enhancing the antitumoral activity of oncolytic viruses. The field of ‘Aptamers’ is growing tremendously in recent years for the targeted delivery and specific treatment. The literature on the aptamer’s conjugation with OVs for tumor treatment is limited and a decade old (authors also stated in lines 358/359). So, I am doubtful about the necessity of reviewing very old research. I request authors to state the intentions of this review and objectives clearly in both abstract and last paragraph of Introduction. Overall, the review is written well and focused to the subject. However, the reviewer thinks that addressing/supplementing following points would enhance the overall quality.  

1)      Supplementing section 2 and 3 with a list of OVs and/or Aptamers approved/undergoing clinical testing for cancer treatment would enhance the overall readability.

2)      Although the review is focused mainly on the use of Aptamers for enhancing OVs anticancer activity, discussing the use of small molecules in combination with OVs for cancer treatment (mutual benefits) in section-2 would elevate the content. (see: 10.3390/cancers13143386) and epigenetic modulators (see:  10.3390/cancers13112761)

3)      Like Figure-1, adding a figure describing SELEX technology and aptamer’s potential model of action (macromolecular protein complexes) would ease to understand section-3.

4)      Please discuss and cite Zamay TN et al. findings discussing DNA-aptamer for targeting Vimentin for cancer therapy (DOI: 10.1089/nat.2013.0471).

5)      In section 4, add a subsection discussing the application of aptamer-conjugated bacteria for enhanced tumor biotherapy (some works by Geng et al. 10.1038/s41467-021-26956-8)

6)      Section-4.1 needs to be elaborated with more relevant literature discussing Aptamer-conjugated polymeric particulates as targeted delivery in AML/gliomas/renal/prostrate/ and colon cancers. See 10.1016/j.ijpharm.2014.12.035; 10.1016/j.jmmm.2013.05.036; 10.1016/j.biomaterials.2011.07.004; 10.1007/s12274-014-0619-4)

7)      The findings by Muharemagic D et al. (ref.37) are interesting. Please elaborate a bit more in section 4.3.1 with discussion of results.

8)      The position of Table-1 in 4.3.3. (for cryoprotection) is not justified.

9)      Expand Table-1 with adding a column with the tumor type (breast/liver/lung/etc.) the listed aptamers tested and found effective.

10)   Though section-2 briefly discussed some limitations, adding a section focused on the ‘Safety aspects’ of these oncolytic viruses (toxicity, environmental shedding, mutation and reversion to wildtype virus) at the end of section-4 would benefit the readers to understand the pros and cons with the OV-mediated anticancer therapy.

11)   Having a paragraph or table describing the properties of both aptamers and OVs as separately and the need or benefits with their conjugation (for cancer and non-cancer diseases) enhancer reader understanding of concepts.

Best wishes,

Author Response

Dear Reviewers and Editors,

Thank you for giving us the opportunity to submit a revised draft of our manuscript titled “Aptamers enhance oncolytic viruses’ antitumor efficacy” to Pharmaceutics. We appreciate the time and effort that You and the reviewers have dedicated to providing your valuable feedback on our manuscript. We are grateful to the reviewers for their insightful comments on my paper.

Please find below our response to reviewer’s comments.

Comments from Reviewer â„–2

The review by Dymova A. et al. describes the current progress with the use of aptamers for enhancing the antitumoral activity of oncolytic viruses. The field of ‘Aptamers’ is growing tremendously in recent years for the targeted delivery and specific treatment.

  1. The literature on the aptamer’s conjugation with OVs for tumor treatment is limited and a decade old (authors also stated in lines 358/359). So, I am doubtful about the necessity of reviewing very old research.

Response: We agree with this and have incorporated your suggestion throughout the manuscript. Indeed, not much research has been done in this area, and they were done about ten years ago. In connection with the spread of SARS-CoV-2 infection, new studies of the interaction of binding aptamers with viral particles have appeared, devoted to reducing the viral load on the body or detecting viral particles. We have added links to highlight new areas related to the interaction of this virus and the aptamer, as well as to increase the number of recent publications. Also we changed the subtitle according to the text content.

  1. I request authors to state the intentions of this review and objectives clearly in both abstract and last paragraph of Introduction.

Response: Thank you for this suggestion. Corrected.

  1. Overall, the review is written well and focused to the subject. However, the reviewer thinks that addressing/supplementing following points would enhance the overall quality.

1)      Supplementing section 2 and 3 with a list of OVs and/or Aptamers approved/undergoing clinical testing for cancer treatment would enhance the overall readability.

Response: The main purpose of this review was precisely to show the possible prospects for combined use of aptamers and oncolytic viruses. In this article, we have described aptamers and oncolytic viruses that are FDA-approved or locally approved in specific countries for use in clinical practice. We also indicated the number of clinical trials listed on the clinicaltrial.gov website. Similar tables, neatly and scrupulously compiled, are available in recent reviews, and it would be more ethical to simply insert a sentence to mention them, which we did - we've added most relevant sentences and links.

2)      Although the review is focused mainly on the use of Aptamers for enhancing OVs anticancer activity, discussing the use of small molecules in combination with OVs for cancer treatment (mutual benefits) in section-2 would elevate the content. (see: 10.3390/cancers13143386) and epigenetic modulators (see:  10.3390/cancers13112761)

Response: Corrected. We added several sentences.

3)      Like Figure-1, adding a figure describing SELEX technology and aptamer’s potential model of action (macromolecular protein complexes) would ease to understand section-3.

Response: Corrected. We added the figure 2 describing the SELEX method.

4)      Please discuss and cite Zamay TN et al. findings discussing DNA-aptamer for targeting Vimentin for cancer therapy (DOI: 10.1089/nat.2013.0471).

Response: We discussed and cited this manuscript.

5)      In section 4, add a subsection discussing the application of aptamer-conjugated bacteria for enhanced tumor biotherapy (some works by Geng et al. 10.1038/s41467-021-26956-8)

Response: Corrected. We added the sentence to the section 4 about aptamer-conjugated bacteria (ApCB).

6)      Section-4.1 needs to be elaborated with more relevant literature discussing Aptamer-conjugated polymeric particulates as targeted delivery in AML/gliomas/renal/prostrate/ and colon cancers. See 10.1016/j.ijpharm.2014.12.035; 10.1016/j.jmmm.2013.05.036; 10.1016/j.biomaterials.2011.07.004; 10.1007/s12274-014-0619-4)

Response: We agree with the reviewer. We have added related links.

7)      The findings by Muharemagic D et al. (ref.37) are interesting. Please elaborate a bit more in section 4.3.1 with discussion of results.

Response: Corrected. We have expanded this subsection by discussing this article in more detail.

8)      The position of Table-1 in 4.3.3. (for cryoprotection) is not justified.

Response: Corrected. This is a summary table for the entire section, so we changed the sentence to “We have summarized all the information provided in Section 4.3. Aptamers in-creased the antitumor efficacy of an oncolytic virus, in Table 1.”.

9)      Expand Table-1 with adding a column with the tumor type (breast/liver/lung/etc.) the listed aptamers tested and found effective.

Response: The articles mentioned in Table 1 show us the possibility of using aptamers to increase the efficiency of virus binding to the target, protection against neutralizing antibodies, and demonstration of cryoprotective properties. Only in the article (doi:10.1021/ja903857f) can we talk about a specific nosology (acute T-cell leukemia). In the article (doi:10.1021/nn2006927), the authors used aptamers against a heteroaryldihydropyrimidine structure, chosen as a representative drug-like molecule with no cross reactivity with mammalian or bacterial cells, in other articles this was not tested, since it was not the goal of the authors.

10)   Though section-2 briefly discussed some limitations, adding a section focused on the ‘Safety aspects’ of these oncolytic viruses (toxicity, environmental shedding, mutation and reversion to wildtype virus) at the end of section-4 would benefit the readers to understand the pros and cons with the OV-mediated anticancer therapy.

Response: Thank you for pointing this out. We added the text and the related links to the manuscript, but to the section -2 (Oncolytic viruses), not to the section -4 (Applications of aptamers).

11)   Having a paragraph or table describing the properties of both aptamers and OVs as separately and the need or benefits with their conjugation (for cancer and non-cancer diseases) enhancer reader understanding of concepts.

Response: Corrected. We added the sentences to conclusion section.

We highlighted (yellow highlighted text) all changes made when revising the manuscript to make it easier for the Editors to give a prompt decision on manuscript. We thank the reviewer for his thoughtful and thorough review and believe that his contribution was invaluable. We thank the editors for considering our work for publication.

Yours faithfully,

Dr. Maya Dymova PhD, Senior researcher

Laboratory of Biotechnology

Institute of Chemical Biology and Fundamental Medicine SB RAS

(ICHBFM SB RAS) Russian Federation, 630090, Novosibirsk, Lavrentjev av., 8

tel: 00 7 9134764012

Reviewer 3 Report

A comprehensive review of  merging the fields f  oncolytic viruses and aptamers that will be very useful for the community  on both the aptamer and oncology sides.  I would recommend to  add some information on the cell  targeting of  oncolytic viruses.  The English language needs some careful reviewing  to improve grammar and clarity 

Author Response

Dear Reviewers and Editors,

Thank you for giving us the opportunity to submit a revised draft of our manuscript titled “Aptamers enhance oncolytic viruses’ antitumor efficacy” to Pharmaceutics. We appreciate the time and effort that You and the reviewers have dedicated to providing your valuable feedback on our manuscript. We are grateful to the reviewers for their insightful comments on my paper.

Please find below our response to reviewer’s comments.

 Comments to reviewer â„–3

A comprehensive review of merging the fields of oncolytic viruses and aptamers that will be very useful for the community on both the aptamer and oncology sides.

  1. I would recommend to add some information on the cell targeting of oncolytic viruses.

Response: Thank you for this suggestion. We added the sentences describing the cell targeting of oncolytic viruses.

  1. The English language needs some careful reviewing to improve grammar and clarity.

Response: We reviewed the final text with a native English-speaking colleague with an individual Grammarly.com account.

We highlighted (yellow highlighted text) all changes made when revising the manuscript to make it easier for the Editors to give a prompt decision on manuscript. We thank the editors for considering our work for publication.

Yours faithfully,

Dr. Maya Dymova PhD, Senior researcher

Laboratory of Biotechnology

Institute of Chemical Biology and Fundamental Medicine SB RAS

(ICHBFM SB RAS) Russian Federation, 630090, Novosibirsk, Lavrentjev av., 8

tel: 00 7 9134764012

Reviewer 4 Report

Manuscript pharmaceutics-2082669 reviews the utility and prospects of aptamers as an enhancer of anti-tumor activity of oncolytic viruses.

This review enables the readers to understand the developments in oncolytic virotherapy, and the concept of this review is meaningful for the clinical interest.

1.           In the section on oncolytic viruses, I recommend the authors explain the mechanism of action of oncolytic viruses in detail.

2.           The authors mentioned the approved oncolytic viruses for cancer treatment in section 2. The authors should add information about the genetically engineered oncolytic herpes simplex virus type 1, teserpaturev, approved by the Pharmaceuticals and Medical Devices Agency.

3.           I recommend the authors add the schema of the outline of SELEX.  

Author Response

Dear Reviewers and Editors,

Thank you for giving us the opportunity to submit a revised draft of our manuscript titled “Aptamers enhance oncolytic viruses’ antitumor efficacy” to Pharmaceutics. We appreciate the time and effort that you and the reviewers have dedicated to providing your valuable feedback on our manuscript. We are grateful to the reviewers for their insightful comments on my paper. Please find below our response to reviewer’s comments.

Comments to reviewer â„–4

Manuscript pharmaceutics-2082669 reviews the utility and prospects of aptamers as an enhancer of anti-tumor activity of oncolytic viruses. This review enables the readers to understand the developments in oncolytic virotherapy, and the concept of this review is meaningful for the clinical interest.

  1. In the section on oncolytic viruses, I recommend the authors explain the mechanism of action of oncolytic viruses in detail.

Response: Corrected. We added the sentences describing the action of oncolytic viruses.

  1. The authors mentioned the approved oncolytic viruses for cancer treatment in section 2. The authors should add information about the genetically engineered oncolytic herpes simplex virus type 1, teserpaturev, approved by the Pharmaceuticals and Medical Devices Agency.

Response: Thank you for this suggestion. We added the information about Teserpaturev / G47Δ (Delytact®)

  1. I recommend the authors add the schema of the outline of SELEX.

Response: Corrected. We added the figure 2 describing the SELEX technology.

We highlighted (yellow highlighted text) all changes made when revising the manuscript to make it easier for the Editors to give a prompt decision on manuscript. We thank the editors for considering our work for publication.

Yours faithfully,

Dr. Maya Dymova PhD, Senior researcher

Laboratory of Biotechnology

Institute of Chemical Biology and Fundamental Medicine SB RAS

(ICHBFM SB RAS) Russian Federation, 630090, Novosibirsk, Lavrentjev av., 8

tel: 00 7 9134764012

Round 2

Reviewer 1 Report

authors addressed reviewer concerns

Reviewer 2 Report

Dear Authors,

Thanks for incorporating reviewer suggestions in the revised version of your manuscript. Overall, the revised manuscript looks in good shape with more relevant literature.

Best wishes,